# Plant Metabolic Engineering by Multigene Stacking: Synthesis of Diverse Mogrosides

**DOI:** 10.3390/ijms231810422

**Published:** 2022-09-09

**Authors:** Jingjing Liao, Tingyao Liu, Lei Xie, Changming Mo, Xiyang Huang, Shengrong Cui, Xunli Jia, Fusheng Lan, Zuliang Luo, Xiaojun Ma

**Affiliations:** 1The Artemisinin Research Center, Institute of Chinese Materia Medica, China Academy of Chinese Medical Sciences, Beijing 100700, China; 2College of Horticulture, Shenyang Agricultural University, Shenyang 110866, China; 3Institute of Medicinal Plant Development, Chinese Academy of Medical Sciences, Peking Union Medical College, Beijing 100193, China; 4Guangxi Crop Genetic Improvement and Biotechnology Lab, Guangxi Academy of Agricultural Sciences, Nanning 530007, China; 5Guangxi Key Laboratory of Plant Functional Phytochemicals and Sustainable Utilization, Guangxi Institute of Botany, Guangxi Zhuang Autonomous Region and Chinese Academy of Sciences, Guilin 541006, China; 6Guilin GFS Monk Fruit Corp, Guilin 541006, China

**Keywords:** mogrosides, natural sweetener, plant chassis, multigene assembly, synthetic biology

## Abstract

Mogrosides are a group of health-promoting natural products that extracted from *Siraitia grosvenorii* fruit (Luo-han-guo or monk fruit), which exhibited a promising practical application in natural sweeteners and pharmaceutical development. However, the production of mogrosides is inadequate to meet the need worldwide, and uneconomical synthetic chemistry methods are not generally recommended for structural complexity. To address this issue, an in-fusion based gene stacking strategy (IGS) for multigene stacking has been developed to assemble 6 mogrosides synthase genes in pCAMBIA1300. Metabolic engineering of *Nicotiana benthamiana* and *Arabidopsis thaliana* to produce mogrosides from 2,3-oxidosqualene was carried out. Moreover, a validated HPLC-MS/MS method was used for the quantitative analysis of mogrosides in transgenic plants. Herein, engineered *Arabidopsis thaliana* produced siamenoside I ranging from 29.65 to 1036.96 ng/g FW, and the content of mogroside III at 202.75 ng/g FW, respectively. The production of mogroside III was from 148.30 to 252.73 ng/g FW, and mogroside II-E with concentration between 339.27 and 5663.55 ng/g FW in the engineered tobacco, respectively. This study provides information potentially applicable to develop a powerful and green toolkit for the production of mogrosides.

## 1. Introduction

Recently, the low-sugar diet gradually became an attractive healthy lifestyle, and plant-derived sweeteners have drawn a tremendous amount of attention around the world. Up to now, many natural sweeteners with high potency have been approved by the U.S. Food and Drug Administration (FDA), including D-psicose (isolated from wheat, fig, and raisin) [1], *Siraitia grosvenorii* fruits extracts (mogrosides) [2], and stevia leaf extracts [3]. Notably, mogrosides, a group of sweet cucurbitane-type triterpene glycosides in Luo-han-guo, have an increasing market demand due to their superior sweetness compared to rebaudiana A [4]. Furthermore, at least so far, D-psicose is still not approved to be used as a food additive in China yet. Consequently, mogrosides are the representative of an emerging trend in the global natural sweetener market [5]. Mogrosides are commonly considered to be a mixture of cucurbitane-type triterpene glycosides with different numbers of glucoses, which have all kinds of pharmacological bioactivities including antitumor [6,7], anti-inflammatory [8], anti-diabetic [9,10], antiglycation and antioxidant [11]. For instance, mogroside V (five glucoses attached to mogrol) is an essential component at levels of 5.77–11.75 mg/g DW in the fruits of *Siraitia grosvenorii*. Siamenoside I (four glucoses attached to mogrol) is the sweetest component and has a preferable flavor than other mogrosides. However, the yield is merely at a lower level of 1 mg/g DW in the fruits of *S. grosvenorii* [12,13]. Additionally, in the mature fruits of *Siraitia grosvenorii*, most of which are seeds, account for approximately 70% of dry weight [14]. Thus, the plant extraction method is almost impossible to yield high concentrations of natural mogrosides, which has limited the extent of its marketability. Unfortunately, the complexity of mogrosides in special glycosylation sites makes the application of chemical synthesis extremely challenging. Furthermore, *Siraitia grosvenorii* is distributed only in China, which is limited by a planting environment of high humidity, high temperature and higher diurnal temperature variation. Additionally, to protect designation of origin, its seeds and genetic genes have already been legislated and are prohibited to be taken out of the country. Furthermore, mogrosides are only present in the fruits of Luo-han-guo, which are not detected in the leave, stem, root and seed [15]. This reduces the potential of market competitiveness for mogrosides which are prohibitively expensive and less economical. Hence, a traditional cultivation mode is apparently struggling to meet a surging global market demand of mogrosides, which urgently requires an environmental and efficient toolkit to be developed to produce mogrosides.

Alternatively, major extraordinary progress in synthetic biology has provided tremendous possibility for the mogrosides production in heterologous hosts. Heterologous biosynthesis of natural products in microbial or plant chassis is one of the research hotspots in the field of synthetic biology, which greatly reduces the natural resource damage, environmental pollution and economic benefits [16,17,18,19]. Researchers have published *Siraitia grosvenorii* genome data and mogrosides biosynthesis pathway has been investigated, which offers a solid theoretical basis for the heterologous production of mogrosides [15,20]. Microbial chassis were preferable in large-scale industrial production, which were comparatively well researched [17,21,22]. Several attempts have been made to explore the mogrosides in *Saccharomyces cerevisiae*. For example, mogroside V has been transformed into mogroside III E by β-glucosidases in the yeast [23]. *Dekkera bruxellensis* Exg1p enzyme has been used to hydrolyze mogroside V to siamenoside I, which contained 62.67 ± 3.71% in the fermentation products [24]. However, it has been not reported for the glycosylation resulting in mogroside V, siamenoside I, and mogroside III yet. This probably results from the lack of subcellular organelles required for functional expression of eukaryotic enzymes (e.g., cytochrome P450s), the absence of substrates or poor endogenous metabolic flux towards mogrosides. An alternative approach has potential to produce health-promoting natural products in plant chassis.

In the past decade, *Nicotiana benthamiana* and *Arabidopsis thaliana* are fast-growing and high-biomass model plants that can be used as a vehicle for metabolic engineering and have been a popular approach to improve the production of natural products, such as taxadiene, taxadiene-5α-ol [25], artemisinin [26], artemisinic acid [27], unusual fatty acid [28], triterpene [29], and DHA [30], etc. For example, artemisinin is an effective anti-malarial drug isolated from *Artemisia annua* with the low yields, which cannot meet the high market demand commonly. In 2011, researchers have reconstructed the artemisinin in tobacco, which paves the way for the development of a sustainable plant-based platform for the production of all kinds of secondary metabolites [26]. Additionally, they have a well- defined genetic background and efficient transformation system, which provides a valuable insight into the establishment of a bioreactor for mogrosides in plants. Taking advantage of the elucidated mogrosides biosynthesis pathway, all genes responsible for the biosynthesis have been characterized, and the precursor, 2,3-oxidosqualene commonly exists in plants. As shown in Figure 1, to de novo synthesize the mogrosides, at least 6 encoding genes are required to be transformed into the heterologous plants, including *SgSQE1* (squalene epoxidase), *SgCS* (cucurbitadienol synthase), *SgEPH2* (epoxide hydrolases), *SgP450* (cytochrome P450 monooxygenase), *SgUGT269-1* and *SgUGT289-3* (UDP-glucosyltransferases). In fact, an appropriate multigene stacking strategy and transformation is used to produce mogrosides in heterologous plants.

Over the past two decades, there have been major advances in multigene stacking, including GATEWAY [31], Gibson assembly [32], In-Fusion [33], Cre recombinase/loxP-mediated recombination (TransGene Stacking II, TGS II) [34,35], and suppression thermo-interlaced PCR (STI PCR) [36]. Furthermore, the construction of fusion protein with Internal ribosomal entry sites (IRESs) and 2A peptides can be used to assemble multiple genes [37,38,39]. So far, there are abundant researches in the field of heterologous production with diverse plant chassis. β-carotene-biofortified rice endosperm [40], maize [41], canola [42] and soybean [43], anthocyanin-enriched rice endosperm [34], tomato [44], and maize [39]; ginsenoside aglycone rice [45]; astaxanthin-enriched rice [35] and tomato [46], and antimalarial drug, artemisinin, have been synthesized in *Nicotiana benthamiana* [26] and *Physcomitrella patens* [47]. Recently, multigene assembly strategy mainly focuses on 2-10 genes transformation in plants, and with the increasing number of transgenes, the vector gradually become unstable, thus leading to gene silencing. Therefore, an efficient multigene stacking strategy is important to improve the efficiency and applicability of multigene vector in all kinds of plant species for future research.

Here, in-fusion based gene stacking strategy (IGS) combined with 2A peptides was developed to assemble 6 mogrosides synthase genes and a mogrosides biosynthesis pathway has been engineered in tobacco and *Arabidopsis*. Our study also yielded a green toolkit for the development of multiple plant materials that are sources of mogrosides and have great market potential. This offered an alternative for production of mogrosides in perspective heterologous plants.

## 2. Results

### 2.1. Multigene Expression Vector Construction

Mogrosides biosynthesis pathway has been studied extensively in *Siraitia grosvenorii*, 2,3-oxidosqualene was used as the substrate, which is existed widely in all kinds of plants [15]. To achieve heterologous biosynthesis of mogrosides, at least 6 mogrosides synthase genes were required to be introducing into the heterologous plants. For this reason, we developed an in-fusion based multigene stacking strategy with P2A peptides linker. Firstly, *SgSQE1*, *SgCS*, *SgEPH2*, *SgP450*, *SgUGT269-1* and *SgUGT289-3* were selected and cloned into the PBI121 (Appendix A) driven by AtPD7, AtUBQ10, and CaMV 35S promoters, respectively. Secondly, the sequence of promoter + target gene + terminator was cloned and inserted into the pCAMBIA1300 (Appendix A). After that, single-gene expression cassette and the double-gene expression cassette were combined into the pCAMBIA1300, where the triple-gene expression cassette were generated. Finally, the region of UBQ10:SgUGT269-1:Tmas::35S:SgUGT289-3:Tnos::35S:SgP450 was amplified and ligated with the sequence of SgSQE1:Thsp::35S:SgCS:Tnos::35S:SgEPH2:Tnos by P2A peptides. The multigene vector U22p-SCE was approximately 21.5 kb (from the left border to the right border) (Appendix A). This construct was used to produce mogrosides in the tobacco and *Arabidopsis thaliana*, which provided an alternative toolkit for mogrosides production in *Nicotiana benthamiana* and *Arabidopsis thaliana*. In our study, the amino acid sequence of Gly-Ser-Gly has been introduced in N-terminal of P2A peptides, which improved the splicing efficiency [48]. This strategy highlights the potential of the natural products’ production in many heterologous plants.

### 2.2. Transient Expression Assays

To investigate whether the transformation of a multigene vector can produce the mogrosides, transient expression was developed in *Nicotiana benthamiana*. Six mogrosides synthase genes were introduced and expressed in tobacco using *Agrobacterium* infiltration. HPLC-MS/MS analysis indicated that relatively less mogrol has accumulated in the leaves of tobacco infiltrated with U22p-SCE at 36 h. There was no mogrol found in the WT tobacco leaves. However, no mogrosides have been synthesized in the leaves of tobacco (Appendix A). The reasons for the lack of mogrosides production in transient expression assay are mainly as follows: the transient expression assay has a short reaction time, which may cause the shortage of substrate accumulation. There is another reason to this result which is the stability of multigene expression vector, which has an adverse effect on enzyme activities. Furthermore, the lack of 2,3-oxidosqualene has contributed to limited accumulation of mogrosides in tobacco. In this case, a stable transformation of multigene expression vectors is necessary for confirming the mogrosides production in *Nicotiana tabacum* and *Arabidopsis thaliana* further.

### 2.3. Generation of Transgenic Tobacco and Arabidopsis thaliana

To assess the availability of *Nicotiana benthamiana* and *Arabidopsis thaliana* as plant chassis for mogrosides production, the U22p-SCE plasmid harboring mogrosides synthase genes were transformed into plants via *Agrobacterium tumefaciens* (Appendix A). Sixteen Hyg-resistant tobacco transgenic lines were obtained. The sequences of the *SgSQE1, SgCS*, *SgEPH2*, *SgP450*, *SgUGT269-1*, *SgUGT289-3*, and Hyg genes were isolated from transgenic plants (Figure 2A). All the mogrosides biosynthesis-related genes were detected in six (N16, N22, N30, N32, N45, and N47) out of sixteen transgenic tobacco lines via PCR, which demonstrated that all target genes in the multigene vector were integrated into the tobacco genome. Then, the expression levels of target genes were measured by qPCR. All the target genes were overexpressed in the transgenic tobacco lines; however, no target genes were detected in the WT plants (Figure 2B). These data showed that the mogrosides biosynthesis-related genes were successfully expressed in *Nicotiana benthamiana*. Likewise, to select and generate transgenic *Arabidopsis thaliana* lines containing mogrosides biosynthesis-related genes, *Arabidopsis thaliana* (Col-0) plants were subjected to the *Agrobacterium*-mediated floral dip method (Figure 3A). T0 seeds were selected on MS media including Hyg, and 11 independent transgenic lines were identified by using PCR-based detection (Figure 3B). Subsequently, 7 transgenic lines were selected for RT-PCR analysis (Figure 3C).

The PCR analysis showed that the ratio of positive transgenic plants was 37.5% and 63.6% for tobacco and *Arabidopsis*, respectively, which were relatively lower. Sometimes, gene silencing and gene loss occurred in the plant transformation. This was probably due to the existence of redundant repetitive sequences and that the T-DNA fragment was large [49]. To avoid the adverse effect on the gene expression level and enzyme catalytic efficiency in heterologous plants, a different promoter and terminator may attempt to be introduced in the multigene vector, and codon optimization is a good choice to improve gene expression. Additionally, antibiotic resistance gene in transgenic plants should be evaluated. Hygromycin B (Hyg) was used as a selectable marker protein in our study, and concentration of hygromycin was highly negatively correlated with the growth of shoots [50]. Explants were severely yellowed and withered due to a high level of hygromycin in the plant transformation. Conversely, lower concentration may cause an extremely difficult screening process for positive plants. Furthermore, to improve the transformation efficiency of a multigene vector, an antibiotic resistance gene should be replaced by kanamycin or glyphosate, which was commonly used in the plant’s genetic engineering.

### 2.4. Heterologous Production of Mogrosides in Transgenic Tobacco and Arabidopsis thaliana

Our goal was to produce and obtain transgenic lines containing mogrosides by multigene transformation, that is, to develop a promising bio-factory for mogrosides production. It is apparent from Figure 4A,B that, compared with mogrosides, there was a majority of MII-E and a few of MIII accumulated in the transgenic tobacco leaves, respectively (Figure 4A,D). The average content of MII-E was 399.27, 3745.54, 5283.57, and 5663.55 ng/g FW, and the average content of MIII was 148.30, 212.92, and 252.73 ng/g FW (Figure 4B,C). It seemed that MI-A was completely catalyzed into the MII-E, but the MII-E was redundant. Surprisingly, HPLC-MS/MS-based analysis indicated that MIII and SI were present in transgenic *Arabidopsis* lines (Figure 5A–C). The average yields of SI were 29.65, 224.57, and 1036.96 ng/g FW in the AA3, AA6, and AU7 transgenic lines, respectively (Figure 6). Moreover, MIII was found only in the AA3 transgenic line, the level of which was 202.75 ng/g FW (Figure 6). MII-E, MIII, and SI production in transgenic tobacco and *Arabidopsis* lines suggests that these lines could represent novel platforms for mogrosides production.

HPLC-MS/MS analysis indicated that a plethora of MIII and a trace of MII-E were detected in the transgenic tobacco lines, and slight MIII and SI accumulated in transgenic *Arabidopsis* lines. Generally, the sweetness of mogrosides mainly depends on the number of glucose moieties and glycosylation sites. Mogroside V with five glucose moieties, the sweetness is approximately 300 times higher than that of sucrose [12,51]. Interestingly, siamenoside I with four glucose moieties was the sweetest component in the mogrosides and is 563 times sweeter than 5% sucrose because of the different glycosylation sites. Although the sweetness of mogroside III (three glucoses attached to mogrol) is tasteless, its intestinal maltase inhibitory effect has been investigated previously [52]. Previous studies commonly obtained siamenoside I, mogroside III or other mogrosides by hydrolyzing glycosidic bonds in the yeast via the biotransformation of mogroside V [23,24,53]. As a result, the strategy in the present study has significant potential for the production of SI in transgenic *Arabidopsis* plants via biotechnology.

## 3. Discussion

In the last few decades, plant synthetic metabolic engineering is representative of an emerging trend in the production of natural compounds [54]. Firstly, it is based on the identification of key synthase genes, gene flux, and regulatory mechanism for biosynthetic pathway of target compounds; then, efficient multigene stacking and transformation are required to establish in the organisms, which is a pivotal step in this research [55]. Generally, one or more exogenous genes often need to be integrated into the heterologous host genome, which is required to develop all kinds of multigene stacking strategies, such as crossbreeding, re-transformation, and co-transformation. However, these strategies are not only time-consuming but also require randomness and blindness. Currently, multigene vector transformation, that is, multiple genes which are linked in a single T-DNA region, has a great advantage over the other traditional methods [56]. All genes are inserted into a chromosomal site in the plant host genomes by *Agrobacterium*-mediated transformation, and inherited as a single unit. Therefore, the loss of exogenous genes can be reduced and can improve the transgenic efficiency. Due to the significant advantages of the multigene vector transformation, this method is more widely used in current synthetic biology, and a variety of multigene stacking strategies have been developed recently [57,58,59,60,61,62,63,64]. Researchers developed a TGS II system for multigene assembly, and astaxanthin, β-carotene, and anthocyanins have been produced in rice endosperm by the transformation of 4-10 structural genes, respectively [34,35]. Protopanaxadiol was successfully synthesized in the transgenic rice by the integration of ginseng *PgDDS* and *PgCYP716A47* genes using GATEWAY [45]. Gibson assembly has been used to engineer the taxadiene and taxadiene-5α-ol biosynthesis pathway in tobacco [25]. These multigene stacking strategies were rarely used for 6 or more genes transformation except for patented system TGS II. In this case, to achieve the de novo synthesis of mogrosides in heterologous plants, a simple and efficient multigene stacking needed to be developed. Therefore, we attempted to assemble 6 mogrosides synthase genes via in-fusion technology and self-cleaving 2A peptides, which provided an alternative toolkit for mogrosides production in *Nicotiana benthamiana* and *Arabidopsis thaliana*. In our study, the amino acid sequence of Gly-Ser-Gly has been introduced in N-terminal of P2A peptides, which improved the splicing efficiency [48]. This strategy highlights the potential of the natural products’ production in many heterologous plants.

The PCR analysis showed the ratio of positive transgenic plants were 37.5% and 63.6% for tobacco and *Arabidopsis*, respectively. The multigene assembly and transformation carrying a large multigene vector cause the unstable expression of multigene in the host plants which are in addition to the existence of redundant repetitive sequences which commonly are the initiator of partial or all foreign genes silencing [49]. These make the transgenic efficiency of a multigene vector relatively lower. To avoid the adverse effect on the gene expression level and enzyme catalytic efficiency in heterologous plants, a different promoter and terminator should be introduced in the multigene vector. Additionally, codon optimization is a good choice to improve gene expression and the antibiotic resistance gene in transgenic plants should be evaluated. Hygromycin B (Hyg) was used as a selectable marker protein in our study, and concentration of hygromycin was highly negatively correlated with the growth of shoots [50]. Furthermore, to improve the transformation efficiency of a multigene vector, an antibiotic resistance gene should be replaced by kanamycin or glyphosate, which was commonly used in the plant’s genetic engineering.

HPLC-MS/MS analysis indicated that a plethora of MII-E and a trace of MII-E were detected in the transgenic tobacco lines, and less MIII and SI accumulated in transgenic *Arabidopsis* lines. Generally, the sweetness of mogrosides differs depending on the number of glycosylation and glycosylation sites. Previous studies have indicated that in mogroside V with five glucose moieties, the sweetness is approximately 300 times higher than that of sucrose [12,51]. Nevertheless, siamenoside I with four glucose moieties was the sweetest component in the mogrosides and is 563 times sweeter than 5% sucrose because of the different glycosylation sites. In addition, the flavor of siamenoside I was better than that of mogroside V [24]. Additionally, mogroside III has three glucose moieties, which is somehow less sweet than mogroside V [65]. Studies found that the extremely low level of siamenoside I in natural extracts has limited the extent of its marketability [25]. Unfortunately, the complexity of siamenoside I in special glycosylation sites makes the application of chemical synthesis extremely challenging. As a result, the strategy in the present study has significant potential for the production of SI in transgenic *Arabidopsis* plants via biotechnology.

Besides, multigene vector U22p-SCE harbouring 6 mogrosides biosynthesis gene has been assembled via in-fusion technology and self-cleaving 2A peptides, and transformed into the tobacco and *Arabidopsis*. Although all candidate genes involved in mogroside biosynthesis pathway were highly expressed in the transgenic lines, mogroside V was not detected in the transgenic lines. There are several possible explanations for this result. Firstly, a possible explanation for these results may be the lack of an adequate precursor accumulation in the tobacco and *Arabidopsis*. To reduce the consumption of substrates, there was an introduction of various rate-limiting enzymes, such as 3-hydroxy-3-methyl glutaryl coenzyme A reductase (HMGR) and squalene synthase (SQS), which is beneficial to the accumulation of substrates. Previous research has found that overexpression of *PnSS* and *PnHMGR* could significantly improve the accumulation of total saponins in *Panax notoginseng*. Additionally, the yield of co-overexpression of *PnSS* and *PnHMGR* was 3-fold and 1.5-fold higher than the control and *PnHMGR*-overexpressed lines, respectively [64]. Therefore, the introduction of rate-limiting enzymes can facilitate the yield of mogrosides in the plant hosts. Another possible explanation for this might be the protein interaction and localization in transgenic plants, which have impacted greatly on the catalytic efficiency of enzyme. We need to attempt to predict the signal peptides and localization for target proteins, and compartmentalized engineering can be used to produce mogrosides in heterologous plants.

Together, our study is highly important because it marks the success of the transformation of mogroside biosynthesis genes in tobacco and *Arabidopsis*, which offers a perspective on producing plant materials that could serve as a source of multiple mogrosides. This strategy offers an excellent alternative to the traditional extraction methods, which is high-valued toolkit for the production of mogrosides. Furthermore, this study provided a necessary theoretical foundation for mogrosides production in heterologous plants.

## 4. Materials and Methods

### 4.1. Plant Material, Chemicals, and Strains

*Nicotiana benthamiana* and *Arabidopsis thaliana* (ecotype Columbia-0) were used for transformation assay. The binary plasmid PBI121 and pCAMBIA1300 were used for multigene vector construction. *E. coli* competent cells including DH5α and XL10-Gold (WeidiBio, Shanghai, China) were used in this study, and *Agrobacterium tumefaciens* competent cell GV3101 (WeidiBio, Shanghai, China) was used for multigene vector transformations.

HPLC-grade methanol, acetonitrile, and formic acid were obtained from Fisher (Emerson, IA, USA). ClonExpress II One Step Cloning Kit, ClonExpress MultiS One Step Cloning Kit, ClonExpress Ultra One Step Cloning Kit, ClonExpress Ultra One Step Cloning Kit, HiScript Ⅲ 1st Strand cDNA Synthesis Kit (+gDNA wiper), and Taq Pro Universal SYBR qPCR Master Mix were obtained from Vazyme Biotech Co., Ltd. (Nanjing, China). Ultrapure RNA Kit (DNase I) was obtained from CWBIO. Co., Ltd. (Beijing, China). Plant genomic DNA kits, restriction enzymes, and DNA Marker were purchased from TianGen Biotech Co., Ltd. (Beijing, China). KOD One PCR master Mix was obtained from TOYOBO Biotech Co., Ltd. (Shanghai, China). The PBI121 and pCAMBIA1300 plasmids were stored in the laboratory. Mogrosides standards, including mogrol, mogroside II-E (MII-E), mogroside III (MIII) and siamenoside I (SI) were purchased from Chengdu Must Bio-Technology Co., Ltd. (Sichuan, China). Other chemical reagents were obtained from Beijing Chemical Corporation (Beijing, China) unless otherwise specified.

### 4.2. Multigene Stacking

The coding genes of *SgSQE1* (squalene epoxidase)*, SgCS* (cucurbitadienol synthase), *SgEPH2* (epoxide hydrolases), *SgP450* (cytochrome P450 monooxygenase), *SgUGT269-1*, and *SgUGT289-3* (UDP-glucosyltransferases) were isolated from *Siraitia grosvenorii* fruit. Ubiquitin 10 (*AtUBQ10*) and serine carboxypeptidase-like *AtSCPL30* (PD7, 456 bp) promoters were cloned from *Arabidopsis thaliana* using KOD One PCR Master Mix. CaMV 35S promoter and NOS terminator were amplified from the PBI121 plasmid, and mannopine synthase (MAS) terminators and heat-shock protein (HSP) 18.2 terminators were chemically synthesized by GENEWIZ (Suzhou, China).

The strategy of the multigene vector construction in this study is shown in Appendix A. Firstly, *SgCS*, *SgEPH2*, *SgP450*, and *SgUGT289-3* were subcloned into the *Bam*HI and *Sac*I sites of PBI121 vector with CaMV 35S promoter and the NOS terminator to generate 35S:SgCS:Tnos, 35S:SgEPH2:Tnos, 35S:SgP450:Tnos, and 35S:SgUGT289-3:Tnos using a ClonExpress II One Step Cloning Kit. Simultaneously, AtPD7:*SgSQE1*:Thsp and UBQ10:SgUGT269-1:Tmas were constructed using a ClonExpress MultiS One Step Cloning Kit. Secondly, PD7:SgSQE1:Thsp and 35S:SgCS:Tnos were fused into the pCAMBIA1300 at the *Eco*RI and *Hin*dIII sites to generate PD7:SgSQE1:Thsp::35S:SgCS:Tnos. Similarly, another double-gene expression cassette, UBQ:SgUGT269-1:Tmas::35S::SgUGT289-3:Tnos was constructed. Thirdly, PD7:SgSQE1:Thsp::35S:SgCS:Tnos and 35S:SgEPH2:Tnos, UBQ10:SgUGT269-1:Tmas::35S:SgUGT289-3:Tnos and 35S:SgP450:Tnos were ligated into the *Eco*RI/*Hind*III sites of the pCAMBIA1300 plasmid, respectively. Finally, UBQ10:SgUGT269-1:Tmas::35S:SgUGT289-3:Tnos::35S:SgP450 (6.1 kb) and SgSQE1:Thsp::35S:SgCS:Tnos::35S:SgEPH2:Tnos (6.5-kb) were ligated by a 2A peptide linker from porcine teschovirus (amino acid sequence is GSGATNFSLLKQAGDVEENPGP) (P2A) via a ClonExpress Ultra One Step Cloning Kit. The multigene expression vector U22p-SCE was assembly. All primers used for multi-gene vector construction are listed in the Appendix A.

### 4.3. Transient Expression Assay

To analyze the availability of multigene expression vector, multigene vector U22p-SCE was introduced into *Agrobacterium tumefaciens* strain GV3101 for transient expression assay in *Nicotiana benthamiana*. The *Agrobacterium* strain with U22p-SCE was cultured in LB medium (25 mg/L rifampicin and 50 mg/L kanamycin) at 28 °C. Then, cells were centrifuged for 15 min at 5000× *g* and resuspended in buffer with 10 mM 2-(N-morpholino) ethanesulfonic acid (MES), 10 mM MgSO_4_, and 200 μM acetosyringone (AS) to the OD_600_ = 0.6. They were kept in the dark about 2-4 h. The strains were infiltrated into the leaves of *Nicotiana benthamiana* using a needleless syringe. Non-infiltrated plants were as the negative control. The assay was performed independently six times.

### 4.4. Agrobacterium-Transformation in Tobacco

Tobacco transformation was performed according to the leaf disc method, with slight modifications. In short, it was primarily necessary to aseptically cut the leaves of one-month-old *Nicotiana benthamiana* plants for the leaves to be used as explants. These leaf explants were pre-cultivated in MS media consisting of 3% sucrose, 0.7% agar, 1 mg/L 6-BA, and 0.2 mg/L naphthylacetic acid (NAA) for 2 d in the dark at 25 °C. Subsequently, leaf explants of *Nicotiana benthamiana* were inoculated with multigene expression vectors via *Agrobacterium tumefaciens* strain GV3101. Next, the explants were transferred to cocultivation media (MS media including 3% sucrose, 0.7% agar, 1 mg/L 6-BA, 0.2 mg/L NAA, and 100 μM AS) for 3 d in the dark at 25 °C. The explants were incubated on regeneration media (MS media including 3% sucrose, 0.7% agar, 1 mg/L 6-BA, 0.2 mg/L NAA, 400 mg/L cefotaxime, and 7 mg/L Hyg) at 25 °C under a 16 h/8 h (light/dark) photoperiod. When the shoots were approximately 2-3 cm in length, rooting media (MS media including 3% sucrose, 0.7% agar, 0.2 mg/L NAA, 400 mg/L cefotaxime, and 7 mg/L Hyg) were used to cultivate the regenerated shoots. The resistant plants were identified by PCR for the 6 candidate genes and Hyg-resistant gene using specific primers (Appendix A). Wild-type (WT) plant was used as a negative control.

### 4.5. Arabidopsis thaliana Transformation

For *Arabidopsis thaliana* transformation, the Col-0 ecotype was used. The multigene vector U22p-SCE was introduced into the *Agrobacterium tumefaciens* strain GV3101 through the freeze-thaw method, and then the floral-dip method involving *Agrobacterium* harboring the multigene expression vector was used as previously described [66]. To screen for the transgenic lines, T0 seeds were harvested and grown on selective media including Hyg (30 mg/L), and then the T1 transgenic lines were grown in a greenhouse under nonstress conditions for further study. The resistant plants were detected by PCR using specific primers (Appendix A). In addition, genomic DNA from WT plants was a negative control.

### 4.6. Gene Expression Level Analysis

Total RNA was extracted from 0.1 g transgenic plant leaves using a CWBIO RNA extraction kit. The first-strand cDNA was s reverse transcribed from 1 μg of total RNA via HiScript Ⅲ 1st Strand cDNA Synthesis Kit (+gDNA wiper). The relative gene expression level of *SgSQE1, SgCS*, *SgEPH2*, *SgP450*, *SgUGT269-1*, and *SgUGT289-3* were measured by real-time quantitative PCR (qRT-PCR) using Taq Pro Universal SYBR qPCR Master Mix. An ABI CFX96^TM^ Real-Time System (Waltham, Massachusetts, USA) with the following reaction procedure: 95 °C for 30 s; 40 cycles of 95 °C for 3 s, 55 °C for 10 s. *Nbactin* (Accession number: XM_016619439) was internal control. This assay was conducted for multiple technical replicates. All the primers used for qRT-PCR are listed in Appendix A.

For the gene expression level analysis of *Arabidopsis*, Reverse Transcription PCR (RT-PCR) was performed according to the protocol. PCR detection with equivalent cDNA as template was used to determine the gene expression level of *SgSQE1*, *SgCS*, *SgEPH2*, *SgP450*, *SgUGT269-1*, and *SgUGT289-3* using the specific primers (Appendix A) via KOD One PCR Master Mix. *Atactin* (Accession number: NM_001338359) was used as positive control. WT plants was the negative control. The gene expression of transgenes in *Arabidopsis* were analyzed with DNA gel electrophoresis.

### 4.7. HPLC-MS/MS Analysis of Mogrosides

The quantitative analysis of mogrosides was performed as previously described with the HPLC-MS/MS method with minor modification [67]. Mogrosides were extracted from 4 g transgenic tobacco leaves and 5 g *Arabidopsis* leaves using 80% methanol by ultrasound extraction. The extraction conditions were as follows: 40 kHz frequency for 1 h at room temperature, and then centrifuged at 5000× *g* for 20 min. After, the collected supernatant was concentrated with nitrogen and diluted in 1 mL methanol.

A 4500 QTRAP LC-MS/MS (AB SCIEX, Toronto, ON, Canada) with an Agilent Poroshell 120 SB C18 column (100 mm × 2.1 mm, 2.7 µm) were used for HPLC-MS/MS analysis. The mobile phase was 0.1% formic acid-water (A) and acetonitrile (B) with a flow rate of 0.2 mL/min. The HPLC condition of mogrosides analysis was as following: 20% B for 0 min; 23% B for 3–5 min; 40% B for 18 min; 20% B for 18.01–20.10 min. For mogrol analysis, 20% B for 0 min; 30% B for 0.5 min; 88% B for 2–4 min, and 20% B for 5.50–8.00 min. The column effluent was monitored by mass spectrometry with electrospray ionization in positive mode and the multiple reaction monitoring (MRM) scanning was employed for quantification. The optimized mass spectrometric conditions and parameters were shown in Table 1. In the current analysis conditions, good linearity was achieved within the investigated ranges (50–5000 ng/mL) for MII-E, MIII, and SI. Typical equations for the standard curves of MII-E, MIII, and SI were y = 173.79x + 4960 (r² = 0.9998), y = 193.16x + 5616.2 (r² = 0.9997), and y = 48.64x + 75.78 (r² = 0.9976), respectively. Therefore, the contents of mogrosides in transgenic plants were calculated by the corresponding equation.

### 4.8. Data Analysis

SPSS 16.0 statistics program (IBM Co., Armonk, NY, USA) was used to analyze the data in this study. All graphs were illustrated by Origin 2019b (OriginLab Co., Northampton, MA, USA). Three biological and technical replicates were conducted in all experiments, respectively.

## Figures and Tables

**Figure 1 ijms-23-10422-f001:**
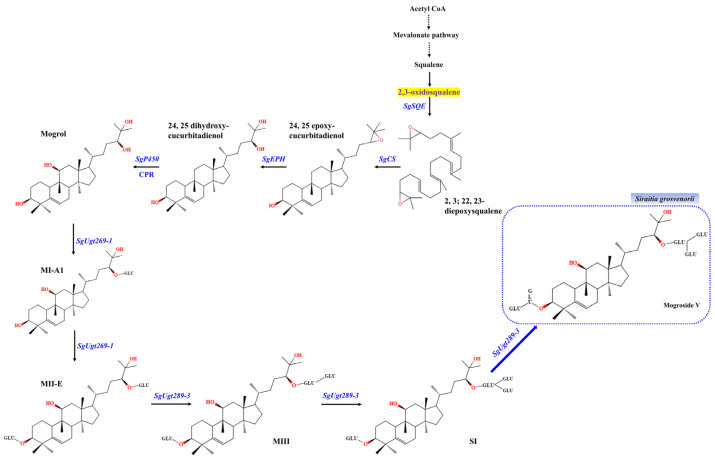
The Flow Chart of mogrosides biosynthetic pathway in *Siraitia grosvenorii* fruits. Mogrosides synthase genes which were transformed in this study are marked in blue, including *SgSQE1*, squalene epoxidases; *SgCS*, curbitadienol synthase; *SgEPH2*, epoxide hydrolases; *SgP450*, cytochrome P450 mono-oxygenase; *SgUGT269-1* and *SgUGT289-3*, UDP-glucosyltransferases; The substrate, 2,3-oxidosqualene is shown with a yellow background. MI-A1, mogroside I-A1; MII-E, mogroside II-E; MIII, mogroside III; SI, siamenoside I; MV, mogroside V (in the dotted bordered rectangle).

**Figure 2 ijms-23-10422-f002:**
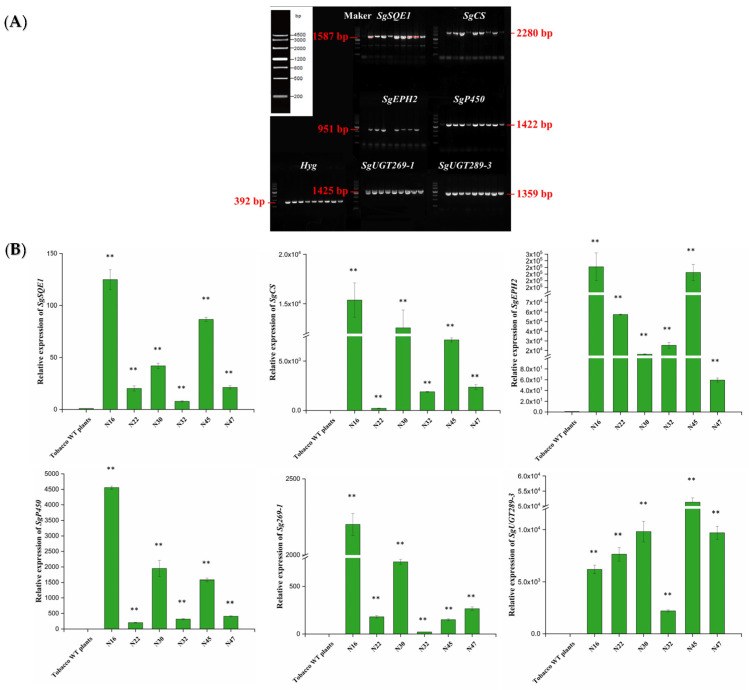
Molecular analysis of transgenic tobacco lines. (**A**) PCR-based analysis of the transgenic tobacco lines. The lanes from left to right represent the WT, N16, N22, N30, N31, N32, N45, N46, N47, and N48. An image of the DNA marker (4.5 kb) is shown in the upper-left corner of the figure. (**B**) Relative expression level analysis of 6 mogrosides biosynthesis genes in transgenic tobacco lines (N16, N22, N30, N32, N45, N47). The *Nbactin* is used as an internal control. Expression of tobacco WT plants was set to 1. The data are presented as the mean values ± SDs, *n* = 3 biologically independent samples, ** represents significant difference at *p* < 0.01 (LSD test).

**Figure 3 ijms-23-10422-f003:**
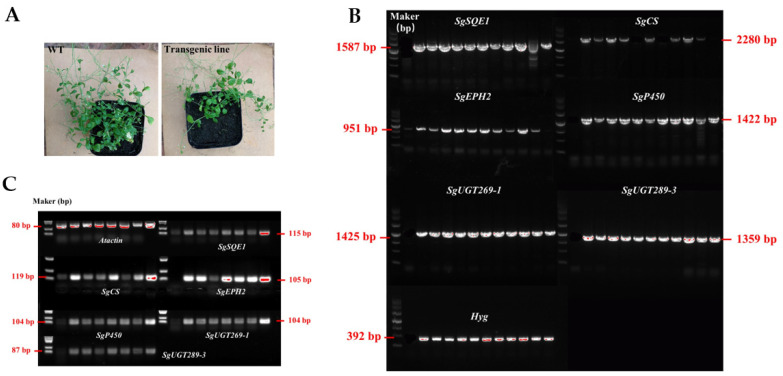
Molecular analysis of transgenic *Arabidopsis* lines. (**A**) *Arabidopsis* WT plants and transgenic lines. (**B**) PCR-based analysis of transgenic *Arabidopsis* lines. The lanes from left to right represent the WT, AA3, AA5, AA6, AA7, AU6, AU7, AU8, AU10, AU11, AU12, and AU13. DNA marker (4500 bp) is used in the figure. (**C**) RT-PCR detection of 6 mogrosides biosynthesis genes in the WT and transgenic *Arabidopsis* lines AA3, AA5, AA6, AU7, AU10, AU11, A12 (from left to right). The *Atactin* is used as an internal control.

**Figure 4 ijms-23-10422-f004:**
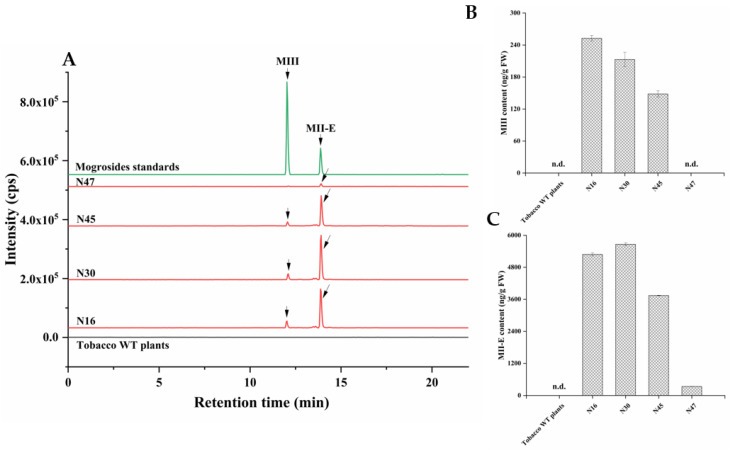
Production of mogrosides in transgenic tobacco lines. (**A**) HPLC-MS/MS analysis of MIII and MII-E in transgenic tobacco lines. (**B**) Accumulation of MIII in transgenic tobacco lines. (**C**) Accumulation of MII-E in transgenic tobacco lines. n.d., not detected. The data are presented as the mean values ± SDs, *n* = 3 biologically independent samples. The black arrows indicate the peak of MIII and MIIE. (**D**) Full-scan product ion of MIII and MII-E.5.0 × 10^4^.

**Figure 5 ijms-23-10422-f005:**
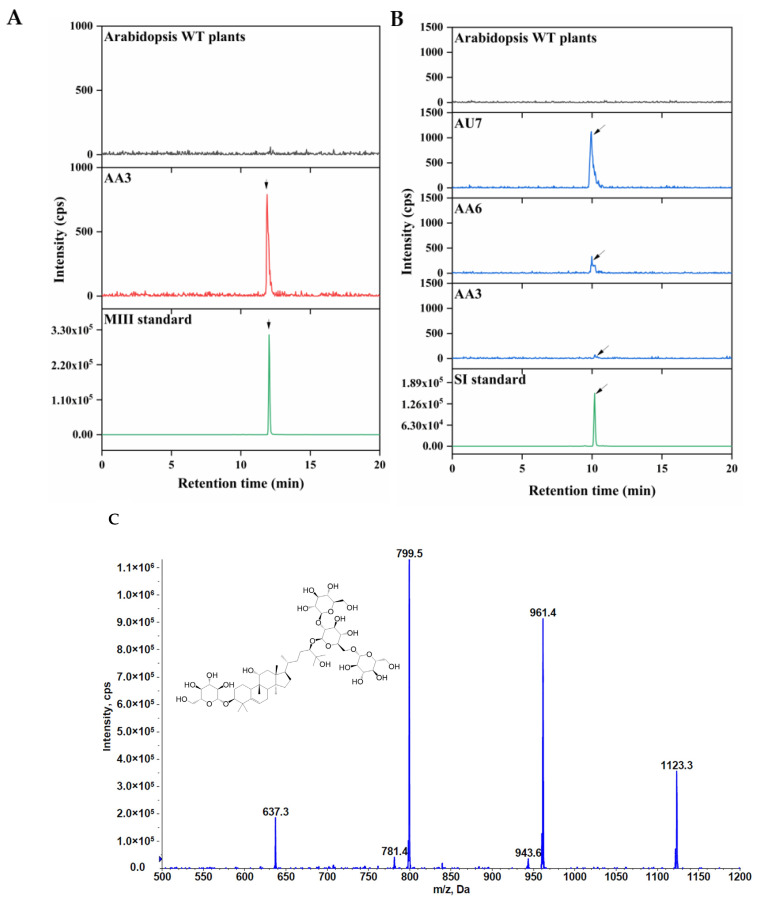
Production of mogrosides in transgenic *Arabidopsis* lines. (**A**) HPLC-MS/MS analysis of MIII in transgenic *Arabidopsis* lines. (**B**) HPLC-MS/MS analysis of SI in transgenic *Arabidopsis* lines. The black arrows indicate the peak of MIII and SI. (**C**) Full-scan product ion of SI.

**Figure 6 ijms-23-10422-f006:**
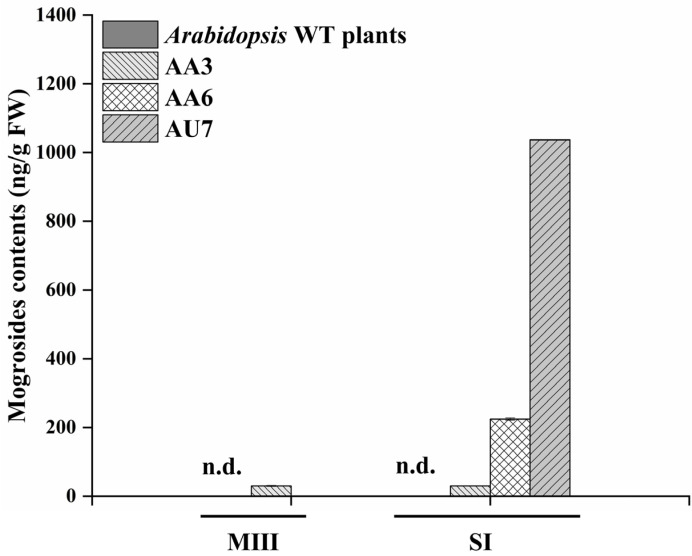
Accumulation of mogrosides in transgenic *Arabidopsis* lines AA3, AA6, and AU7. n.d., not detected. The data are presented as the mean values ± SDs, *n* = 3 biologically independent samples.

**Table 1 ijms-23-10422-t001:** HPLC-MS/MS parameters.

Analytes	Molecular Formula	Retention Time (min)	Production (*m*/*z*)	DP (V)	CE (eV)
**SI**	C_54_H_92_O_24_	9.62	1123.6/961.6 1123.6/799.2	−220	−75
**MIII**	C_48_H_82_O_19_	11.86	961.6/799.4 961.6/637.3	−170	−70
**MII-E**	C_42_H_72_O_14_	13.82	799.5/637.5 799.5/475.5	−170	−65
**Mogrol**	C_30_H_52_O_4_	4.01	459.3/441.2 459.3/423.3	80	20
**MS parameters**		**Mogrosides**		**Mogrol**	
**Ion mode**		Negative		Positive	
**Source temperature (℃)**		550		550	
**Ionization voltage (V)**		−4500		5500	
**GS1 (psi)**		55		18	
**GS2 (psi)**		55		20	
**CUR (psi)**		20		20	
**CAD**		Medium		Medium	
**Dwell time (ms)**		100		200	
**EP (V)**		−10		10	
**CXP (V)**		−15		10	

## Data Availability

The authors confirm that all data in this experiment are available in the main text and Appendix A.

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
