# Peer review of "Plant Metabolic Engineering by Multigene Stacking: Synthesis of Diverse Mogrosides"

_ijms, 2022, doi:10.3390/ijms231810422_

Round 1
Reviewer 1 Report
The manuscript entitled: “Plant metabolic engineering by multigene stacking: synthesis of diverse mogrosides” contains some information of potential interest to readers. To improve the manuscript, I would suggest only a few corrections/supplementation. For more details, please see below
The paper is well written and organized. Especially the introduction is a very pleasant read.
1. The Abstract section is well described but, in this part, report the main methods (e.g HPLC-MS/MS) of their work.
2. In introduction part, the authors should highlight the novelty of their work with respect to previous reviews.
3. The authors need to provide the accession number of the beta actin gene that was used as a reference in RT-PCR.
4. Figure 2A, 3B & 3C. The gel picture of marker, as well as the product size of each gene, should be included in the image and try to remove red mark and improve the quality of picture.
5. Figure. 2B needs appropriated statistics.
6. Check Figure 3 caption and legends were correctly mentioned.
7. Figure 4. Metabolite profiling, the author may include standard product m/z ions in overall picture to make it easier for the readers.
8. The author should conclude about how the quantification was done for either single or multiple point injection of standard compounds as well as the concentration of standard injection. If it is possible, try to include the calculation formulae in current manuscript.
9. After reading the Ms, several questions arise that are definitively not discussed, such as What it would happen if all genes with rate-limiting enzyme HMGR and SQS are together of in pairs?
10. The authors need to indicate the number of biological and technical replicates when conducting all the experiments?
I have some concerns about how the results are presented. Your findings are the story you'll tell in response to the research questions you've answered. Why did the author include quantification data in Fig 4 MIII if Fig 5 SI was not included?
The authors are strongly encouraged to improve the paper and to resubmit!
Author Response
Response to comments:
- The main methods of their work have been added in the Abstract section.
- In introduction part, we have highlighted the novelty of previous reviews.
- The accession number of the beta actin gene have been added in this manuscript.
- The gel picture of Figure 2A, 3B & 3C have been modified according to your suggestions and improve the quality of picture.
- 2B have been modified.
- Figure 3 caption and legends have been correctly mentioned in the manuscript.
- The MS/ MS spectrometry of standard products have been added in the Figure 4D and 5C.
- The content of mogrosides was calculated using standard curve method, which the equation has been added in the manuscript.
- The discussion has improved, such as the discussion about the rate-limiting enzyme HMGR and SQS, and so on.
- The number of biological and technical replicates have been added in the Data analysis section
- The quantification data of MII and SI was presented in the Figure 6.
Reviewer 2 Report
The manuscript by Liao J et. al 2022 focused on the metabolic engineering of Nicotiana benthamiana and Arabidopsis thaliana for the production of mongrosides via the expression of mogrosides synthases from Siraitia grosvenorii. Based on the findings obtained, the authors succeeded in demonstrating the functional expression of the biosynthetic genes in both plant systems. The manuscript requires revision prior to its acceptance for publication in this journal.
Major comments
1. Labels in Fig 2 and Fig. 3 (No 3D labeling) need to be revised. For Figure 1, please the full name of the compounds in the caption.
2. Sentences on Line 109-206 and Line 309-318 are exactly the same. Please paraphrase and add more citations with regard to the possible reasons of the low transgenic percentage in other plant systems. Could also elaborate the expression of any of the 6 synthase genes in microbial systems in the Discussion section.
3. SgUGT289-3 is an important enzyme that catalyzed the formation of the putative final products including Mogroside V which could not be detected in the engineered plants. It would be great if the authors could perform the expression of this enzyme in microbial hosts (E. coli/yeast) and then incubate the purified enzyme with siamenoside I as the substrate. This would confirm the proposed biocatalysis of this enzyme for the biosynthesis of mogroside V as suggested in Figure 1.
4. In the Discussion section, the authors should elaborate more on the advantages of the In-Fusion Multigene Stacking strategies especially enabling the biosynthesis of complex natural products in plant chassis. For instance, how long would it take to complete the assembly?
Minor comments
Please revise/modify the language (some typos/missing words) for the following sentences;
Line 134- PBI121 (Figure S2A). driven by
Line 138- Finally, we the region of
Line 258- Maltase inhibitory effect? Please clarify this sentence.
Line 309- may attempt to be introduced
Author Response
Response to comments:
- Labels in Fig 2 and Fig. 3 have been revised. The full name of the compounds has been added in the caption of Figure 1.
- Line 309-318 has been revised. And the low transgenic percentage has been discussed. And our experiment focus on the plant chassis, in this case, we reduced the discussion about the heterologous synthesis in the microbial systems.
- The microbial synthesis of MV have been completed by our group and Itkin’s group (Itkin, M.; Davidovich-Rikanati, R.; Cohen, S.; Portnoy, V.; Doron-Faigenboim, A.; Oren, E.; Freilich, S.; Tzuri, G.; Baranes, N.; Shen, S.; et al. The biosynthetic pathway of the nonsugar, high-intensity sweetener mogroside V from Siraitia Grosvenorii. Proc Natl Acad Sci USA 2016, 113, E7619–E7628. doi:10.1073/pnas.1604828113). SI was used as substrate to synthesize the MV in the coli. However, we paid more attention to the SI, which the sweetest component in the mogrosides, in the manuscript, we were not mentioned the biocatalysis of mogroside V methods.
- Discussion section has been revised, and advantages of the Multigene Stacking strategies have been discussed.
- Line 134- PBI121 (Figure S2A). driven by has been revised.
- Line 138- Finally, we the region of has been revised.
- Line 258- intestinal has been added.
- Line 309- may attempt to be introduced has been revised.